# 7,12-Dimethylbenz(a)anthracene as a Model for Ovarian Cancer Induction in Rats

**DOI:** 10.3390/biology14010073

**Published:** 2025-01-14

**Authors:** Ana Carolina Ciseski Gonçalves, Katia Candido Carvalho, José Antônio Orellana Turri, Ricardo dos Santos Simões, Jesus Paula Carvalho, Luiz Fernando Ferraz da Silva, José Maria Soares Júnior, Edmund Chada Baracat

**Affiliations:** 1Faculdade de Medicina de Bauru, Universidade de São Paulo, Bauru 17012-901, SP, Brazil; anaciseski@usp.br; 2Laboratório de Ginecologia Estrutural e Molecular (LIM 58) do Departamento de Obstetrícia e Ginecologia, Faculdade de Medicina da Universidade de São Paulo, São Paulo 17012-901, SP, Brazil; carvalhokc@gmail.com (K.C.C.); antonioturri@usp.br (J.A.O.T.); 3Disciplina de Ginecologia, Departamento de Obstetrícia e Ginecologia, Hospital das Clínicas da Faculdade de Medicina da Universidade de São Paulo, São Paulo 17012-901, SP, Brazil; ricardo.simoes@hc.fm.usp.br (R.d.S.S.); jesuspaulacarvalho@gmail.com (J.P.C.); 4Departamento de Patologia, Faculdade de Medicina FMUSP, Universidade de São Paulo, São Paulo 17012-901, SP, Brazil; burns@usp.br

**Keywords:** ovarian cancer, DMBA, animal model

## Abstract

Ovarian cancer has the highest mortality rate of all gynecological cancers, given the lack of specific symptoms early on. This lack leads to a late diagnosis, and the dissemination of the disease outside the ovary greatly hinders its treatment and cure. The mechanisms of ovarian carcinogenesis are not yet fully understood. To increase knowledge, experimental animal models have recently been developed and reported in the literature. Among the chemicals used in these models for developing epithelial ovarian cancer, 7,12-dimethylbenz(a)anthracene (DMBA) is the most widely used. However, the advantages and disadvantages of employing this substance remain unclear. The present study is a systematic review of articles available in the main scientific databases focusing not only on the induction time, dose, and administration route but also on specific rat strains for inducing rat ovarian tumors using DMBA. We conclude that in most cases, ovarian carcinogenesis can be studied using DMBA as an inducer in rat and mouse models.

## 1. Introduction

Ovarian carcinomas (OCs) are ranked third among all gynecological cancers in terms of mortality rates, according to data from 2020 [1]. Thousands of women globally are affected, with 325,000 new cases and 207,000 deaths worldwide recorded in 2022 alone [2]. Although comprising only 2.5% of cancers in women, OCs account for 5% of deaths [3]. In Brazil, OC is estimated to be among the 10 most common cancer types in women, with 7310 new cases per year predicted for the 2023–2026 triennium by the Instituto Nacional do Cancer (National Cancer Institute of the Health Ministry of Brazil) [4]. 

From a histological perspective, OC can be divided into five types: high-grade serous, endometrioid, clear cell, mucinous, and low-grade serous carcinoma. High-grade serous carcinoma may develop in the fallopian tube epithelium, and it frequently has TP53 mutations, while low-grade serous carcinoma commonly presents KRAS and BRAF [5]. The 5-year survival rate is nearly 29% for patients with stage III or IV and greater than 92% for patients with stage I carcinoma [6]. Because we lack effective imaging and biomarkers for the early diagnosis of ovarian carcinomas, the prognosis has remained unchanged since around 1980 [7]. Given the need for further studies on ovarian cancers, precancerous lesions, and early carcinomas, focusing mainly on their occurrence, development, and imaging, researchers have proposed several experimental models of study.

Chemically induced OC in animal models highlights the processes of oncogenesis, development, invasion, and metastasis [8]. The chemical compound DMBA (7,12-dimethylbenz[a]anthracene) is widely used in animal models to induce breast carcinoma, ovarian adenocarcinomas, and skin cancer. Moreover, several studies have shown that DMBA-induced ovarian adenocarcinomas are genetically similar to human ovarian adenocarcinomas [9]. Figure 1 shows the steps and current possibilities for DMBA use in rat models.

Nevertheless, there are still many controversies regarding the ideal experimental conditions for the use of DMBA in rat models of ovarian carcinogenesis [8,9]. The present study aimed to find the best model for investigating ovarian carcinogenesis and to determine the most appropriate DMBA dose, administration route, and induction time.

## 2. Materials and Methods

A systematic literature review was conducted according to criteria established by the PRISMA (Preferred Reporting Items for Systematic Reviews and Meta-Analyses) consensus [10].

The electronic databases Medline and Embase were searched with no restrictions on the year of publication. Article selection was completed in May 2024. Only those studies published in English, Portuguese, Italian, and French and retrieved as full texts were deemed eligible.

To retrieve the articles, a search strategy was organized combining the following terms: (9,10-dimethyl-1,2-benzanthracene OR 7,12-dimethylbenzanthracene OR 7,12 dimethylbenzanthracene OR 7,12-dimethylbenz(a)anthracene OR DMBA) AND (Ovary OR Ovarian neoplasm OR Ovarian cancer) in PubMed/Medline. Research in Embase was performed using the following terms: ‘7,12 dimethylbenz[a]anthracene’/exp AND ‘ovary tumor’/exp.

The studies using DMBA to induce ovarian cancer and those showing the results of DMBA used in isolation were included in our data analysis. Studies researching breast carcinogenesis and the genetic pattern of ovarian carcinogenesis, those using aromatase inhibitors, those investigating hormone patterns associated with ovarian carcinogenesis, and those evaluating DMBA-induced carcinogenesis associated with other patterns were excluded. Two authors (A.C.C.G. and J.M.S.J.) independently conducted the search for articles by evaluating titles and abstracts based on the inclusion and exclusion criteria. When there was a disagreement about the inclusion of an article in the review, a third author (R.d.S.S.) was consulted.

The data extracted from the qualified studies were the publication year, country of origin, first author’s name, rat strain, DMBA dose, administration route, time lapse between the DMBA application and tumor development, the success rate of tumor induction, and mortality rate. Additionally, the risk of bias was assessed based on the ROBBINS-I tool; two authors (A.C.C.G. and J.M.S.J.) worked together on the risk-of-bias summary, with no automation tool used in the process.

A meta-analysis was conducted on DMBA as an inducer of ovarian tumors and on rat mortality. Groups were formed according to rat strains, routes of DMBA administration, induction times, and doses. The meta-analysis was performed by calculating the proportion of positive cases relative to the total number of cases with respect to each rat strain, dose, administration route, and induction time in both the animals for treatment and the control animals and then comparing the results of these two groups.

Forest plot graphics correspond to the synthesis of results grouped from similar studies. The effect size of each study represents the effectiveness of each treatment in inducing ovarian cancer in rats treated with DMBA, with a 95% confidence interval, compared to the control or comparator group. The red diamond represents the grouped effect size for each type, dose, or method of introducing DMBA. Black dashed lines represent the point of nullity or the absence of effect on efficacy inducing ovarian cancer. The red dashed line represents the central point of the effect size, and the width of the red diamond margins represents the combined confidence interval for each treatment group. Individual studies or red diamonds that cross the black line can be interpreted as lacking efficacy or statistically significant in their effectiveness, as they contain confidence intervals that cross the point of nullity or have an absence of effect.

The I^2^-Higgins heterogeneity test was performed for all outcomes in the meta-analysis, and the fixed or random effects model was used when applicable. For graphical analysis, forest plots were constructed, and the overall and grouped data are presented in the tables. The STATA 16-SE software was used for all analyses, with a 5% level of significance.

## 3. Results

Article searches, identification, and selection were carried out as described in Figure 2. From among the 412 titles and abstracts from PubMed and the 82 articles from Embase, 42 articles were chosen following the inclusion and exclusion criteria, and they were then narrowed down to 16 for the systematic review.

The total number of animals involved in the experiments was 1484, most of which were of the Sprague Dawley and Wistar strains. The mean age of the animals was 45 days (SD = 0.957 days). Animals were exposed to DMBA for an average of 226.5 days (SD = 130 days). The most utilized application methods were cotton soaked in DMBA, an injection under the bursa, ovarian tissue immersed in melted DMBA, and suture with DMBA. The mean success rate of the experiments was 63.6%; that is, approximately 943 animals developed an ovarian tumor following the application of DMBA. The characteristics of these studies are shown in Table 1. Some of the articles included in this study presented more than one method of cancer induction in their results. 

In order to include all these data in this meta-analysis, the articles by X. Y. Yang (2021) [13] and Y. Huang (2012) [19] are cited more than once to analyze different methods of induction separately.

### 3.1. Meta-Analysis Results

A meta-analysis was conducted on studies using DMBA as an inducer of ovarian tumors as well as on rat mortality. For the former, the studies were grouped according to doses, routes of administration, rat strains, and induction times. The meta-analysis was performed by calculating the proportion of positive cases relative to the total number of cases for each rat strain, dose, administration route, and induction time in both the treatment animals and the control animals. We then compared the results of these two groups.

#### 3.1.1. Effectiveness of DMBA in Ovarian Cancer Induction

The meta-analysis determined that DMBA was significantly effective in ovarian cancer induction in all four categories, as detailed below. The overall effect size was the same for all categories: ES = 0.41; 95% CI, 0.31–0.51; *p* < 0.001 (Appendix A).

##### Effectiveness by Dose

The studies were grouped into three categories of DMBA doses: 0.5–0.9, 1–1.9, and 2–3 mg/kg. All doses had a significant ES, indicating that DMBA is a significantly effective inducer of ovarian cancer regardless of the dose used (see Figure 3 and Table 2).

##### Effectiveness by Administration Route

For the statistical analysis of the administration route, the results were also positive. The highest ES was that of epidermic application (ES = 0.60; 95% CI, 0.53–0.67). The other routes also exhibited good results with no statistically significant differences among them. Therefore, in general, all routes of DMBA administration showed good results for the induction of ovarian carcinogenesis (see Figure 4 and Table 3).

##### Effectiveness by Rat Strain

The meta-analysis of the studies found that DMBA was significantly effective in inducing ovarian cancer independent of the rat strain. Most strains had a positive ES that was statistically significant. Such was the case with Fisher, Sprague Dawley, and Wistar rates with B6c3 and C57, which are, therefore, strains that can be used in animal model experiments for studying ovarian carcinogenesis (see Figure 5 and Table 4).

##### Effectiveness by Induction Time

The induction time in the studies ranged from fewer than 110 days to 500 days, and this was categorized for statistical analysis as up to 110 days, up to 180 days, up to 210 days, and up to 500 days. Between 110 and 180 days, the ES was 0.60, 95% CI, 0.34–0.85; between 180 and 210 days, the ES was 0.41, 95% CI, 0.25–0.57; and between 210 and 500 days, the ES was 0.45, 95% CI, 0.25–0.65. These values suggest that the relationship between induction time and an increase in carcinogenesis is not proportional and that extending the study duration does not result in better outcomes. Still, when the induction time is up to 110 days, the ES is 0.17, 95% CI, 0.07- 0.28; this leads to the conclusion that fewer than 110 days seems insufficient for carcinogenic induction (see Figure 6 and Table 5).

#### 3.1.2. Mortality

The meta-analysis of the studies indicated that the mortality rate was always significant, regardless of the doses, administration routes, rat strains, or induction times. Overall, the effect size was the same for all four categories (ES = 0.27; 95% CI, 0.16–0.37; *p* < 0.001) (Appendix A).

##### Mortality by Dose

The mortality rate was significant regardless of the treatment dosage. Overall, the effect size was 0.27; 95% CI, 0.16–0.37; *p* < 0.001 (see Figure 7 and Table 6).

##### Mortality by Route of DMBA Administration

The mortality rate was significant, regardless of the administration route. Overall, the effect size was 0.27; 95% CI 0.16–0.37; *p* < 0.001 (see Figure 8 and Table 7).

##### Mortality by Rat Strain

The mortality rate was significant regardless of the type of rat. Overall, the effect size was 0.27; 95% CI, 0.16–0.37; *p*< 0.001 (see Figure 9 and Table 8).

##### Mortality by Induction Time

The mortality rate was significant regardless of the induction time. Overall, the effect size was 0.27; 95% CI, 0.16–0.37; *p* < 0.001 (see Figure 10 and Table 9).

## 4. Discussion

Dimethylbenzanthracene is a carcinogen only recently used in animal models to research ovarian carcinogenesis. It was already known for its ability to induce tumor initiation, promotion, and progression [9]. Although its action mechanism is not yet fully understood, its direct application in the ovary is known to provoke neoplastic induction in 37% of cases [25]. The application results in hypoplasia and cell destruction, which seem to occur through a connection between DMBA metabolites and DNA, causing the inhibition of Tp53 [26]. Such an alteration is believed to trigger a malignant transformation in rat ovarian epithelial cells, supporting DMBA use in an experimental model since a change in Tp53 is a frequent mutation in human ovarian tumors [12]. Such a model is relevant for mutagenic mechanisms that resemble those of common environmental carcinogens, like the polycyclic aromatic hydrocarbons present in polluted air and smoke [27,28].

We found no systematic review in the literature that standardized a chemical induction of animal carcinogenesis; hence, a strong point of the present study was its aim to select an experimental animal model and specify the animal strain as well as adequate DMBA doses for ovarian carcinogenesis.

Our study possesses other strengths. These include a wide literature search with no restrictions on the year of publication. As a result, more than 400 articles published up to May 2024 were retrieved, encompassing all the experiments related to our objectives. Furthermore, a meta-analysis enabled conclusions based on the included studies. The studies were analyzed using a random effects statistical model, with the I^2^ test revealing high heterogeneity. This should be considered when interpreting the results. Furthermore, the statistical analyses indicated no significant differences in effect size among the groups.

Some limitations exist in the present study. Despite the numerous articles in the literature using DMBA as an inducer of carcinogenesis, many were excluded from this review because they used DMBA in combination with other substances, making it difficult to differentiate the carcinogenic effects of each substance. In addition, even those studies with similar experimental methodologies greatly diversified the presentations of their results, hindering an assessment of all of the varieties of DMBA used.

The ROBINS-I tool of PRISMA 2020 was used to assess the risk of bias in the studies included in this review (Appendix A). The studies were evaluated by two authors (A.C.C.G. and J.M.S.J.) considering the following five categories: R—bias arising from the randomization process; D—bias due to deviations from intended interventions; Mi—bias due to missing outcome data; Me—bias in the measurement of the outcome; and S—bias in the selection of the reported results. No serious risk of bias was identified in the studies. There were only concerns considering Mi, Me, and S. In research by Cai et al. (2019) [8], Tunca J. C. (1985) [11], Seriya S. (1979) [12], Crist K. A. (2005) [18], Nishida T. (2000) [20], Hilfrich J. (1975) [21], Jacobs A. J. (1983) [22], Taguchi O. (1988) [23], and Howell J. S. (1954) [24] there might be biases due to missing outcome data (Mi). In studies by Tunca J. C. (1985) [11], Craig Z. R. (2010) [15], Huang Y. (2012) [19], Nishida T. (2000) [20], Hilfrich J. (1975) [21], and Jacobs A. J. (1983) [22], there were concerns about bias in the measurement of outcomes (Me). Additionally, Cai et al. (2019) [8], Craig Z. R. (2010) [15], Crist K. A. (2005) [18], Huang Y. (2012) [19], Nishida T. (2000) [20], Hilfrich J. (1975) [21], Jacobs A. J. (1983) [22], and Howell J. S. (1954) [24] seemed to have a moderate bias in the selection of reported results (S). Considering the overall bias, there were only concerns about Nishida T. (2000) [20], Hilfrich J. (1975) [21], and Jacobs A. J. (1983) [22], as those studies raised questions in three out of five categories of ROBINS-I.

In consonance with that, the meta-analysis found no high risk of bias. In the analysis using Begg’s test and a funnel plot of all the studies grouped, we determined that there was a low risk of bias (less than 1%) for increased effectiveness, and for mortality, no bias was identified.

The articles selected for this systematic review employed different methods of DMBA application for inducing ovarian carcinogenesis. The use of this substance is relevant, not only because it was used most often but also because it was associated with high rates of ovarian tumors in an animal model.

From the meta-analysis, we detected no statistical difference between DMBA doses, animal strains, or administration routes. We found a significant ES in all studies, meaning that the treatment of animals nearly always experienced a strong effect from the DMBA application compared to the control group. Overall, the results were all similar: regardless of animal strain, administration route, induction time, and dose, the induction of ovarian tumors with DMBA as the carcinogenic inducer was always high.

In every study, only one animal strain was used, and the results varied according to the doses and the administration routes employed.

The animal species used in the experiments were Sprague Dawley, B6C3F1, Fisher 344, and Wistar rats, along with C3H28, C3HeB, FEJ29, and hybrid strains. Statistically, the results did not differ significantly among the varied rat strains. Therefore, the dissimilar results and different types of OC reported by the papers in this review were consequences of the diverse DMBA doses and administration routes.

For example, the Wistar species were tested by Crist et al. (2005) [16], Huang et al. (2012) [17], and Nishida et al. (2000) [18] using different doses and administration routes. Crist et al. (2005) [16] used 0.2 mg of DMBA applied with a silk suture. Huang et al. (2012) [17] used two routes and doses: a suture with 1 mg of DMBA and a cloth strip immersed in 1.5 mg of melted DMBA. Their results showed that a suture with a higher dose of carcinogen increased the percentage of tumors: 38% for 0.2 mg [16] and 46% for 1.0 mg [17]. In the study by Nishida et al. (2000) [18], application through a suture with DMBA yielded an even higher rate (47.5%) despite a much smaller dose (0.0021 mg). The proportion of adenocarcinomas increased according to the dose that was used: 34% in the Crist et al. (2005) study [16] and 56% in the Huang et al. (2012) study [17].

In the experiment using the Sprague Dawley rats, the surgical exposure of the ovaries, followed by the application of cotton soaked in 3.0 mg of DMBA, generated 90% of the tumors. Of these, 87.7% were serous tumors, and 33.1% were SBOTs (serous borderline ovarian tumors). The method employing DMBA-soaked cloth was adopted by Cai et al. (2019) [11] in Sprague Dawley rats and by Huang et al. (2012) [17] in Wistar rats. The former experiment yielded results in 50 to 110 days and produced 90% of tumors, whereas the latter experiment spanned 224 days of induction and produced only 75% of the tumors. The main difference in the results, however, seemed to derive from the administration route since the percentage of ovaries with tumors increased to 75% when a cloth strip immersed in melted DMBA was applied [17], and the percentage of adenocarcinomas was also higher (93.75%). In the study by Nishida et al. (2000) [18], the percentage of nonepithelial tumors was low (10.53%). The method with the best results in those studies was that of the DMBA-soaked cloth, which could be a cloth strip immersed in DMBA [17], gauze soaked in DMBA [9], or cotton soaked in DMBA [11]. The impact of the DMBA administration route was discussed by Huang et al. (2012) [17], who assumed that a suture inevitably causes harm to the ovarian stroma, provoking diverse tumor types besides the epithelial type, such as the granulosa cell tumor. Thus, the suture method does not seem to be adequate in an animal model investigating ovarian carcinogenesis, given that, in humans, the serous tumor is the most prevalent subtype: it accounted for 87.7% of the cases in the Cai et al. (2019) study [11].

The presentation of such results in isolation may suggest that the dose and administration route of DMBA have a significant impact on the outcome. However, the statistical analysis of all the studies found that these two factors were not as influential as they seemed. In other words, when statistically compared, all doses and administration routes had significant ESs.

Among the three categories, there was no single significant difference in ES results and no improvement in results proportional to dose increases, different rat strains, or administration routes. Nevertheless, considering the studies included in this review and the meta-analysis results, the best DMBA dose for carcinogenesis induction is between 0.5 and 3 mg. Furthermore, Sprague Dawley, B6C3F1, Fisher 344, and Wistar rats along with C3H28, C3HeB FEJ, and hybrid rat strains can be successfully used.

Induction time was the only category found to be significantly effective both in individual papers and collectively through statistical analysis. However, a comparison of ESs revealed a reduction in effectiveness when the induction time was fewer than 110 days. Within the 110-to-500-day range, the differences in the number of tumors induced for any period of time were not statistically significant. Thus, we recommend that the induction time in studies of DMBA-induced carcinogenesis is between 110 and 500 days. Evidently, the choice of a particular length of time within this period depends upon factors other than tumor percentages, namely, the researchers’ resources and time availability.

The mortality rate of the animals should be considered to prevent incompatibility with good experimental outcomes. We believe that a good animal model should have a low mortality rate. Therefore, the lower the ES, the better the statistical result. In all categories, the ES was less than 50%, ranging from 16% to 37%.

## 5. Conclusions

Ovarian carcinogenesis is frequently studied using DMBA as an inducer in animal models. Between 0.5 and 3 mg of DMBA can be applied through DMBA-soaked cotton or suture, subcapsular injection, the direct immersion of tissue in DMBA, or the epidermal or intragastric route. Although results may appear before 110 days, the induction time should range between 110 and 500 days. The availability of resources should be considered during the planning stages, and the number of animals used should be calculated for a mortality rate of between 16% and 37%.

## Figures and Tables

**Figure 1 biology-14-00073-f001:**
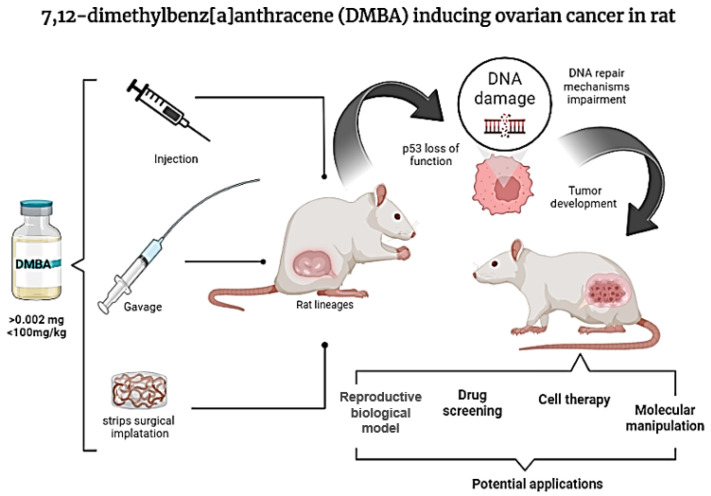
Summary of the current methods and possibilities for DMBA use as an ovarian cancer inductor in rat models. Created with BioRender.com based on the references [5,6,7,8,9].

**Figure 2 biology-14-00073-f002:**
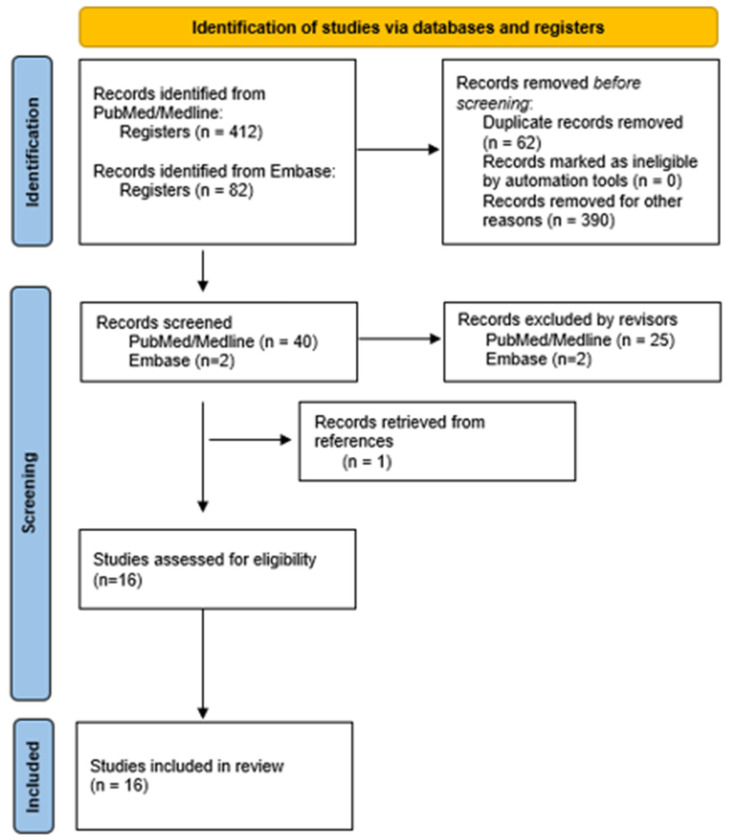
Flowchart of the search for and selection of articles used in the present systematic review.

**Figure 3 biology-14-00073-f003:**
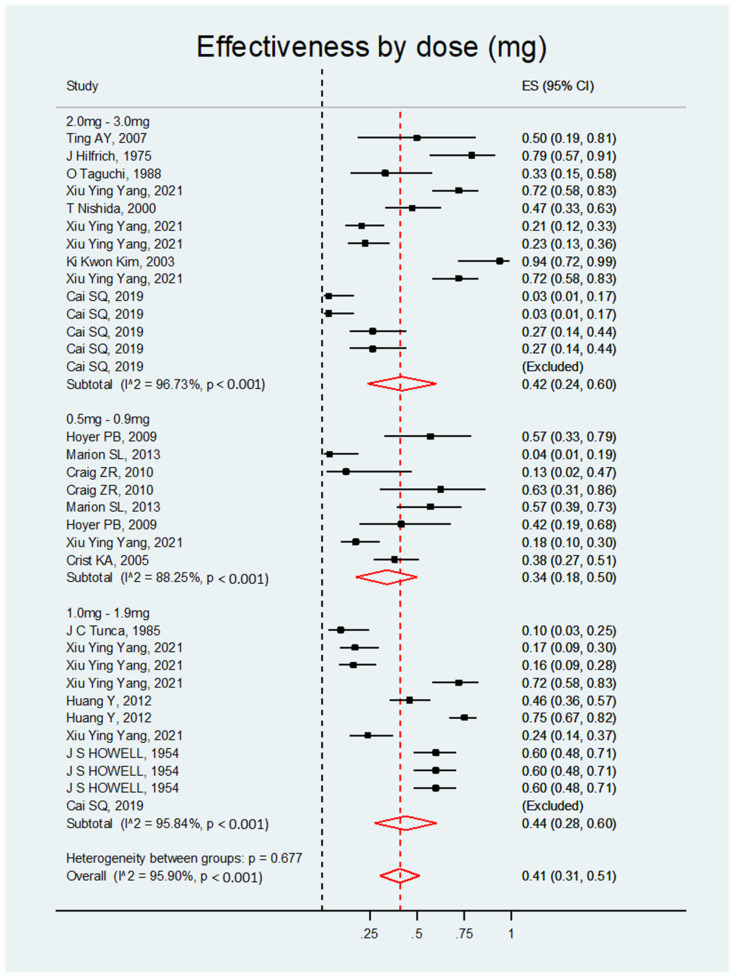
Forest plot of the DMBA effect on ovarian cancer induction in rats grouped by dose.

**Figure 4 biology-14-00073-f004:**
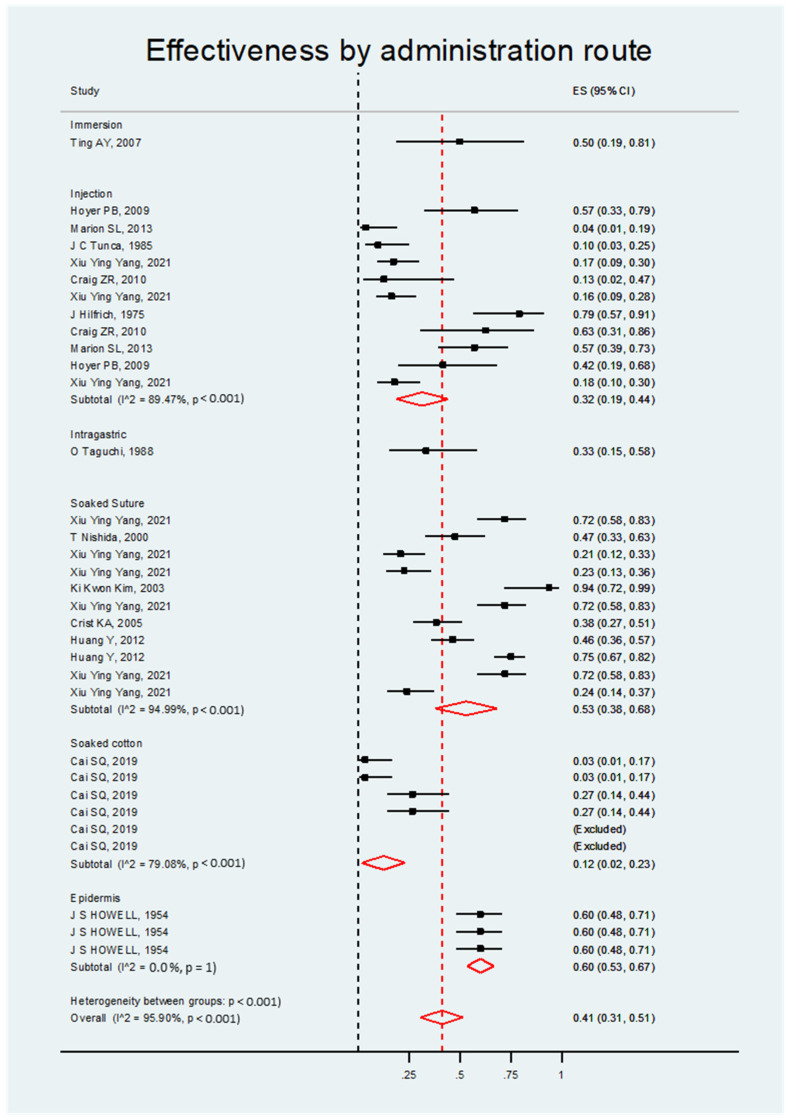
Forest plot of the DMBA effect on ovarian cancer induction in rats grouped by administration route.

**Figure 5 biology-14-00073-f005:**
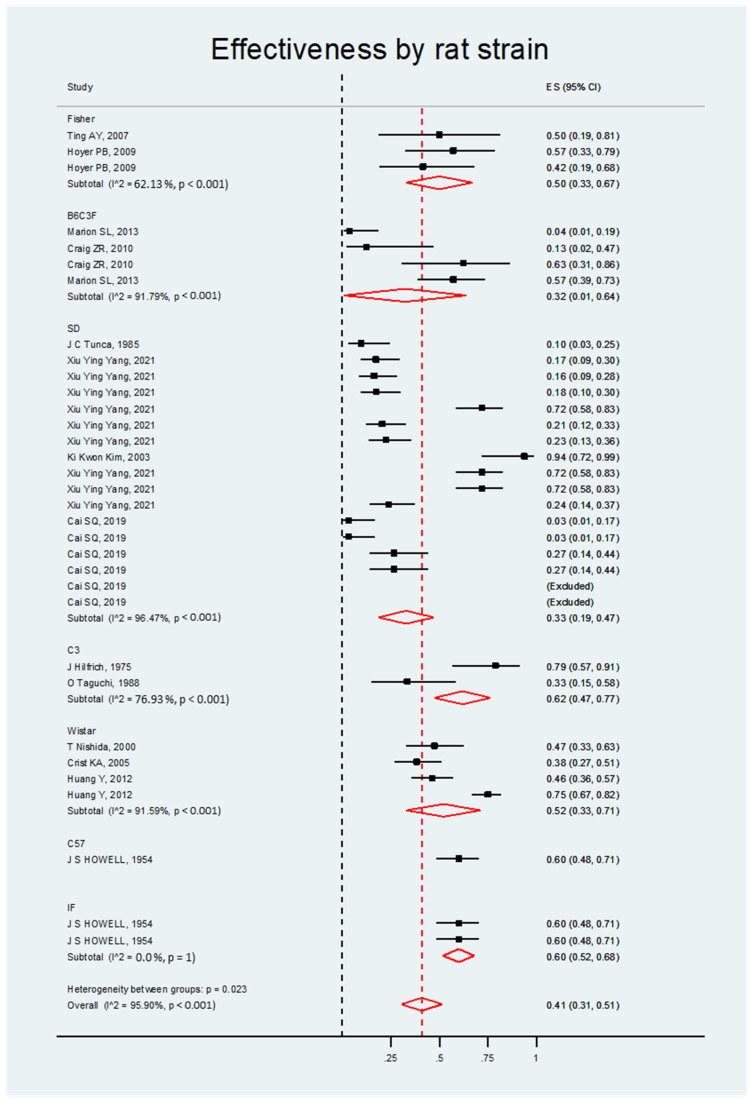
Forest plot of the DMBA effect on ovarian cancer in rats grouped by strain.

**Figure 6 biology-14-00073-f006:**
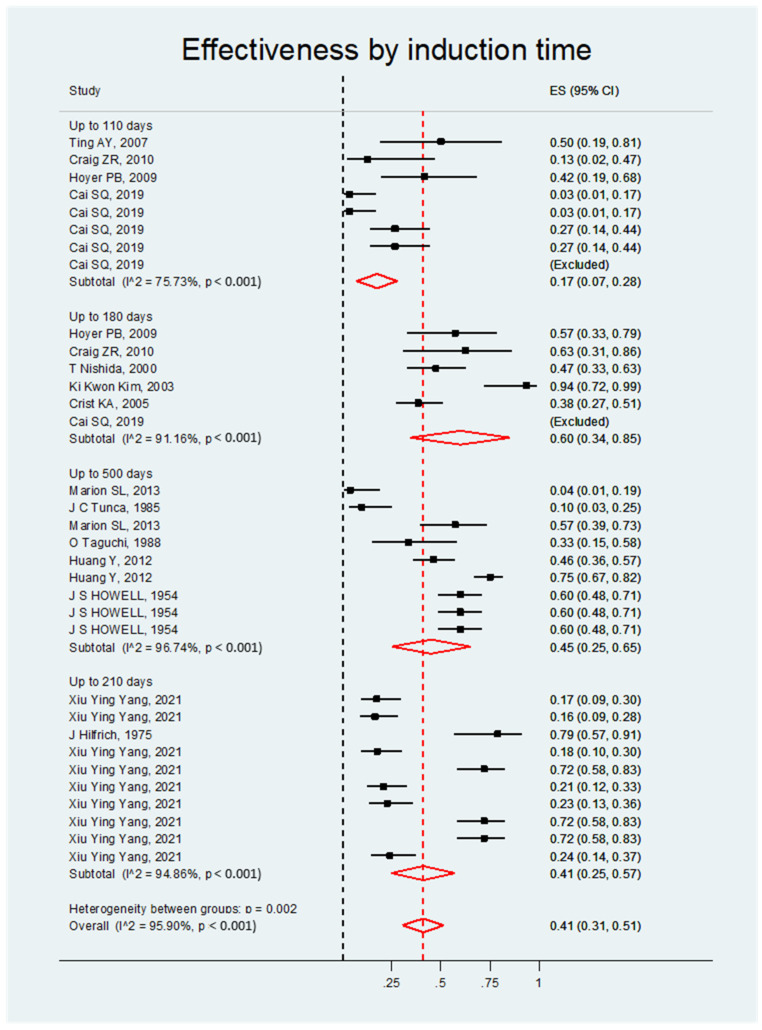
Forest plot of the DMBA effect on ovarian cancer induction in rats grouped by induction time.

**Figure 7 biology-14-00073-f007:**
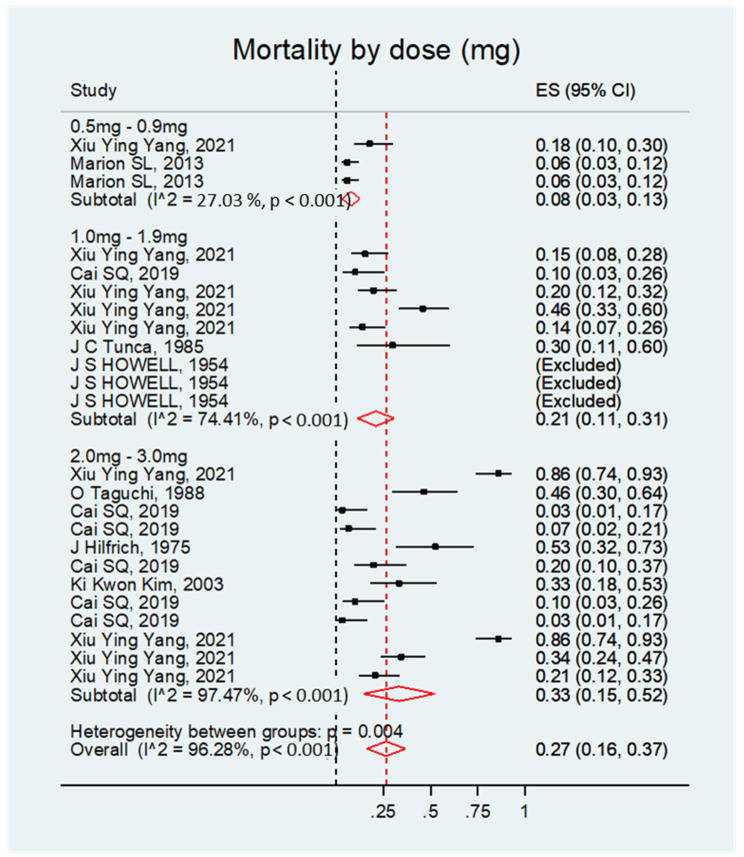
Forest plot of the mortality rate of rats with DMBA-induced ovarian cancer grouped by treatment dose.

**Figure 8 biology-14-00073-f008:**
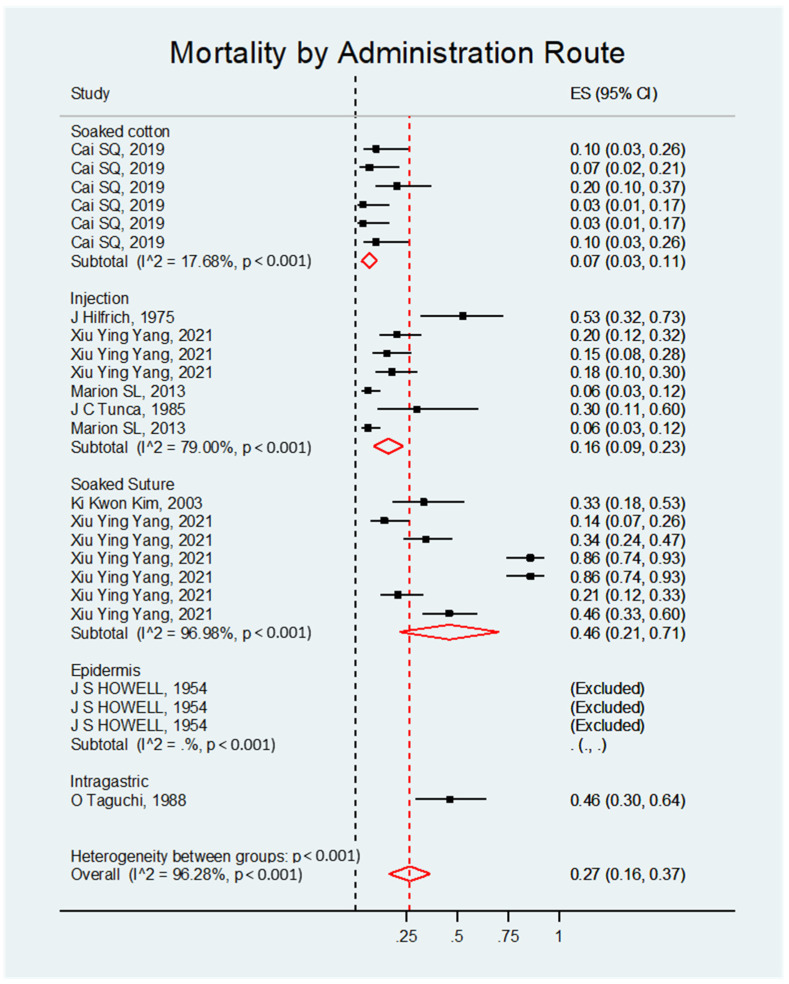
Forest plot of the mortality rate of rats with DMBA-induced ovarian cancer grouped by administration route.

**Figure 9 biology-14-00073-f009:**
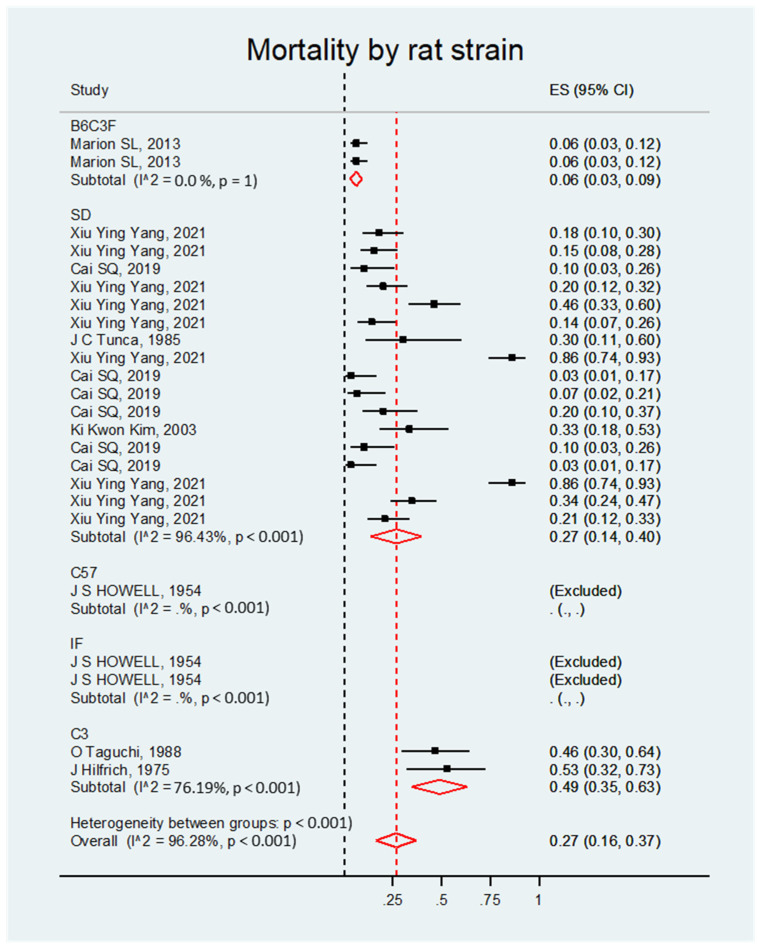
Forest plot of the mortality rate of rats with DMBA-induced ovarian cancer grouped by strain.

**Figure 10 biology-14-00073-f010:**
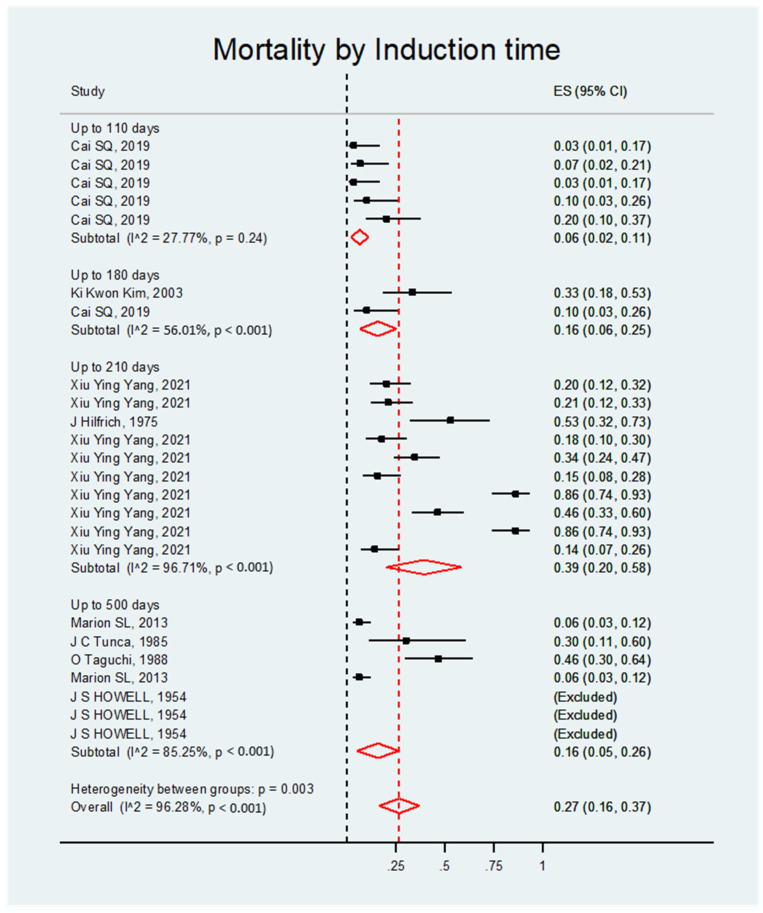
Forest plot of the mortality rate of rats with DMBA-induced ovarian cancer grouped by induction time.

**Table 1 biology-14-00073-t001:** Features extracted from the articles in this study.

Author, Year	Country	Rat Lineage	DMBA Dose (mg)	Administration Route	Time for Tumor Induction (Days)	Success Rate (Live Animals with Tumors)	Mortality
Cai et al. 2019 [9]	China	Sprague Dawley	1–3	Surgical exposure of the ovaries and DMBA-soaked cotton application	70–20–110	Total: 11.1%	0.08%
Kim, K. K. 2003 [10]	South Korea	Sprague Dawley	0.002	Suture with DMBA	140	Total: 93.75%	37.5%
Tunca, J. C. 1985 [11]	USA	Sprague Dawley	1 cm suture soaked in DMBA	Suture with DMBA	407	Total: 10%	3%
Sekiya, S. 1979 [12]	Japan	Sprague Dawley	1.92	Suture with DMBA	-	Total: 48%	
Yang, X. Y. 2021 [13]	China	Sprague Dawley	1, 2, or 3	DMBA-soaked gauze on ovary	60–180	Total: 78%	72.67%
Yang, X. Y. 2021 [13]	China	Sprague Dawley	0.5, 1.0 or 1.5	Subcapsular injection in the ovary	60–180	Total: 75.6%	17.61%
Yang, X. Y. 2021 [13]	China	Sprague Dawley	1, 2, or 3	DMBA-soaked gauze on ovary	60–180	Total: 87.8%	25.16%
Marion, S. L. 2013 [14]	USA	B6C3F1	0.05	Diluted injection under the bursa in the right ovary	210	Total: 31.4%	6%
Craig, Z. R. 2010 [15]	USA	B6C3F1	0.05	Diluted injection under the bursa in the right ovary	90–150	Total on d 90 of induction: 12.5%; Total on d 150 of induction: 57.1%	
Hoyer, P. B. 2009 [16]	USA	Fisher 344	0.1	Diluted injection under the bursa in the right ovary	90–150	Total in 90 d of induction: 42% Total in 150 d of induction: 57%	
Ting, A. Y. 2007 [17]	USA	Fisher 344	2.5 mm region dipped in melted DMBA	Ovarian tissue extraction and immersion in melted DMBA	90–180	30% to 50%	
Crist, K. A. 2005 [18]	UK	Wistar	0.2	Silk suture immersed in melted DMBA for 10 s and air-dried	175	Total: 38%	
Huang, Y. 2012 [19]	China	Wistar	1.5	Cloth strip immersed in melted DMBA	224	Total: 75%	
Huang, Y. 2012 [19]	China	Wistar	1	Suture with DMBA	224	Total: 46%	
Nishida, T. 2000 [20]	Japan	Wistar	0.0021	Suture with DMBA	133–252	Total: 47.5%; Epithelial: 89.47%; Nonepithelial: 10.53%	0
Hilfrich, J. 2975 [21]	Germany	C3H	100 mg/kg	Intravenous injection	203	Total: 78.9%	55.5%
Jacobs, A. J. 1983 [22]	USA	C3HeB FEJ	-	Suture with DMBA	540	Total: 2.8%	
Taguchi, O. 1988 [23]	1988, Japan	Hybrid lineage: C3H/HeMs x 129/J F1	20 mg/kg	Diluted in oil, through digestive tube	360	Total: 33.3%	46.43%
Howell, J. S. 1954 [24]	1954, England	Bonser IF and 2 hybrid lineages (IF x Strong A and IF x C57 Black)	1.25	Drop application on body surface: 4 drops each on ventral and dorsal surfaces	120–210	Total: 60%	0

**Table 2 biology-14-00073-t002:** Meta-analysis of the DMBA effect on ovarian cancer induction in rats grouped by dosage.

Study	ES	95% CI	Heterogeneity (%)
**0.5 mg–0.9 mg**			
Marion SL, 2013 [14]	0.04	0.01–0.19	0.8825
Marion SL, 2013 [14]	0.57	0.39–0.73	
Hoyer PB, 2009 [16]	0.42	0.19–0.68	
Hoyer PB, 2009 [16]	0.57	0.33–0.79	
Craig ZR, 2010 [15]	0.13	0.02–0.47	
Craig ZR, 2010 [15]	0.63	0.31–0.86	
Crist KA, 2005 [18]	0.38	0.27–0.51	
Xiu Ying Yang, 2021 [13]	0.18	0.1–0.3	
**Randomly pooled ES**	**0.34**	**0.18–0.5**	***p* < 0.001**
**1.0 mg–1.9 mg**			
Huang Y, 2012 [19]	0.75	0.67–0.82	0.9538
Huang Y, 2012 [19]	0.46	0.36–0.57	
Tunca JC, 1985 [11]	0.10	0.03–0.25	
Xiu Ying Yang, 2021 [13]	0.72	0.58–0.83	
Xiu Ying Yang, 2021 [13]	0.17	0.09–0.30	
Xiu Ying Yang, 2021 [13]	0.16	0.09–0.28	
Xiu Ying Yang, 2021 [13]	0.24	0.14–0.37	
Howell JS, 1954 [24]	0.60	0.48–0.71	
Howell JS, 1954 [24]	0.60	0.48–0.71	
Howell JS, 1954 [24]	0.60	0.48–0.71	
Cai SQ, 2019 [8]	excluded		
**Randomly pooled ES**	**0.44**	**0.29–0.59**	***p* < 0.001**
**2.0 mg–3.0 mg**			
Cai SQ, 2019 [8]	0.03	0.01–0.17	0.9673
Cai SQ, 2019 [8]	0.03	0.01–0.17	
Cai SQ, 2019 [8]	0.27	0.14–0.44	
Cai SQ, 2019 [8]	0.27	0.14–0.44	
Ting AY, 2007 [17]	0.50	0.19–0.81	
Ki Kwon Kim, 2003 [9]	0.94	0.72–0.99	
Taguchi O, 1988 [23]	0.33	0.15–0.58	
Hilfrich J, 1975 [21]	0.79	0.57–0.91	
Nishida T, 2000 [20]	0.47	0.33–0.63	
Xiu Ying Yang, 2021 [13]	0.72	0.58–0.83	
Xiu Ying Yang, 2021 [13]	0.72	0.58–0.83	
Xiu Ying Yang, 2021 [13]	0.23	0.13–0.36	
Xiu Ying Yang, 2021 [13]	0.21	0.12–0.33	
Cai SQ, 2019 [8]	excluded		
**Randomly pooled ES**	**0.42**	**0.24–0.60**	***p* < 0.001**
**Overall**			
**Randomly pooled ES**	**0.41**	**0.31–0.51**	***p* < 0.001**

**Table 3 biology-14-00073-t003:** Meta-analysis of the DMBA effect on ovarian cancer induction in rats grouped by administration route.

Study	ES	95% CI	Heterogeneity (%)
**Soaked cotton**			
Cai SQ, 2019 [8]	0.03	0.01–0.17	0.7908
Cai SQ, 2019 [8]	0.03	0.01–0.17	
Cai SQ, 2019 [8]	0.27	0.14–0.44	
Cai SQ, 2019 [8]	0.27	0.14–0.44	
Cai SQ, 2019 [8]	Excluded		
Cai SQ, 2019 [8]	excluded		
**Randomly pooled ES**	**0.12**	**0.02–0.23**	***p* = 0.02**
**Peritoneal injection**			
Marion SL, 2013 [14]	0.04	0.01–0.19	0.8947
Marion SL, 2013 [14]	0.57	0.39–0.73	
Hoyer PB, 2009 [16]	0.42	0.19–0.68	
Hoyer PB, 2009 [16]	0.57	0.33–0.79	
Craig ZR, 2010 [15]	0.13	0.02–0.47	
Craig ZR, 2010 [15]	0.63	0.31–0.86	
Tunca JC, 1985 [11]	0.10	0.03–0.25	
Hilfrich J, 1975 [21]	0.79	0.57–0.91	
Xiu Ying Yang, 2021 [13]	0.18	0.10–0.30	
Xiu Ying Yang, 2021 [13]	0.17	0.09–0.30	
Xiu Ying Yang, 2021 [13]	0.16	0.09–0.28	
**Randomly pooled ES**	**0.32**	**0.19–0.44**	***p* = 0.02**
**Organ immersion**			
Ting AY, 2007 [17]	0.50	0.19–0.81	***p* < 0.001**
**Soaked gauze**			
Crist KA, 2005 [18]	0.38	0.27–0.51	0.9452
Huang Y, 2012 [19]	0.75	0.67–0.82	
Huang Y, 2012 [19]	0.46	0.36–0.57	
Ki Kwon Kim, 2003 [9]	0.94	0.72–0.99	
Nishida T, 2000 [20]	0.47	0.33–0.63	
Xiu Ying Yang, 2021 [13]	0.72	0.58–0.83	
Xiu Ying Yang, 2021 [13]	0.72	0.58–0.83	
Xiu Ying Yang, 2021 [13]	0.72	0.58–0.83	
Xiu Ying Yang, 2021 [13]	0.24	0.14–0.37	
Xiu Ying Yang, 2021 [13]	0.23	0.13–0.36	
Xiu Ying Yang, 2021 [13]	0.21	0.12–0.33	
**Randomly pooled ES**	**0.52**	**0.38–0.67**	***p* < 0.001**
**Gavage**			
Taguchi O, 1988 [23]	0.33	0.15–0.58	***p* = 0.01**
**Topic**			
Howell JS, 1954 [24]	0.60	0.48–0.71	-
Howell JS, 1954 [24]	0.60	0.48–0.71	
Howell JS, 1954 [24]	0.60	0.48–0.71	
**Randomly pooled ES**	**0.60**	**0.53–0.67**	***p* < 0.001**
**Overall**			
**Randomly pooled ES**	**0.41**	**0.31–0.51**	***p* < 0.001**

**Table 4 biology-14-00073-t004:** Meta-analysis of the effect of DMBA on ovarian cancer induction in rats grouped by strain.

Study	ES	95% CI	Heterogeneity (%)
**SD**			
Cai SQ, 2019 [8]	0.03	0.01–0.17	0.9625
Cai SQ, 2019 [8]	0.03	0.01–0.17	
Cai SQ, 2019 [8]	0.27	0.14–0.44	
Cai SQ, 2019 [8]	0.27	0.14–0.44	
Ki Kwon Kim, 2003 [9]	0.94	0.72–0.99	
Tunca JC, 1985 [11]	0.10	0.03–0.25	
Xiu Ying Yang, 2021 [13]	0.72	0.58–0.83	
Xiu Ying Yang, 2021 [13]	0.72	0.58–0.83	
Xiu Ying Yang, 2021 [13]	0.72	0.58–0.83	
Xiu Ying Yang, 2021 [13]	0.18	0.10–0.30	
Xiu Ying Yang, 2021 [13]	0.17	0.09–0.3	
Xiu Ying Yang, 2021 [13]	0.16	0.09–0.28	
Xiu Ying Yang, 2021 [13]	0.24	0.14–0.37	
Xiu Ying Yang, 2021 [13]	0.23	0.13–0.36	
Xiu Ying Yang, 2021 [13]	0.21	0.12–0.33	
Cai SQ, 2019 [8]	(Excluded)		
Cai SQ, 2019 [8]	(Excluded)		
**Randomly pooled ES**	**0.34**	**0.2–0.47**	***p* < 0.001**
**B6C3F**			
Marion SL, 2013 [14]	0.04	0.01–0.19	0.9179
Marion SL, 2013 [14]	0.57	0.39–0.73	
Craig ZR, 2010 [15]	0.13	0.02–0.47	
Craig ZR, 2010 [15]	0.63	0.31–0.86	
**Randomly pooled ES**	**0.32**	**0.01–0.64**	***p* = 0.04**
**Fisher**			
Hoyer PB, 2009 [16]	0.42	0.19–0.68	0.9159
Hoyer PB, 2009 [16]	0.57	0.33–0.79	
Ting AY, 2007 [17]	0.50	0.19–0.81	
**Randomly pooled ES**	**0.50**	**0.33–0.67**	***p* < 0.001**
**Wistar**			
Crist KA, 2005 [18]	0.38	0.27–0.51	-
Huang Y, 2012 [19]	0.75	0.67–0.82	
Huang Y, 2012 [19]	0.46	0.36–0.57	
Nishida T, 2000 [20]	0.47	0.33–0.63	
**Randomly pooled ES**	**0.52**	**0.33–0.71**	***p* < 0.001**
**C3**			
Taguchi O, 1988 [23]	0.33	0.15–0.58	-
Hilfrich J, 1975 [21]	0.79	0.57–0.91	
**Randomly pooled ES**	**0.62**	**0.47–0.77**	***p* < 0.001**
**IF**			
Howell JS, 1954 [24]	0.60	0.48–0.71	-
Howell JS, 1954 [24]	0.60	0.48–0.71	
**Randomly pooled ES**	**0.60**	**0.52–0.68**	***p* < 0.001**
**C57**			
Howell JS, 1954 [24]	0.60	0.48–0.71	-
**Randomly pooled ES**	**0.60**	**0.52–0.68**	***p* < 0.001**
**Overall**			
**Randomly pooled ES**	**0.41**	**0.31–0.51**	***p* < 0.001**

**Table 5 biology-14-00073-t005:** Meta-analysis of the DMBA effect on ovarian cancer induction in rats grouped by induction time.

Study	ES	95% CI	Heterogeneity (%)
**Up to 110 days**			
Cai SQ, 2019 [8]	0.03	0.01–0.17	75.73
Cai SQ, 2019 [8]	0.03	0.01–0.17	
Cai SQ, 2019 [8]	0.27	0.14–0.44	
Cai SQ, 2019 [8]	0.27	0.14–0.44	
Hoyer PB, 2009 [16]	0.42	0.19–0.68	
Craig ZR, 2010 [15]	0.13	0.02–0.47	
Ting AY, 2007 [17]	0.50	0.19–0.81	
**Randomly pooled ES**	**0.17**	**0.07–0.28**	***p* < 0.001**
**110 to 180 days**			
Hoyer PB, 2009 [16]	0.57	0.33–0.79	91.16
Craig ZR, 2010 [15]	0.63	0.31–0.86	
Crist KA, 2005 [18]	0.38	0.27–0.51	
Ki Kwon Kim, 2003 [9]	0.94	0.72–0.99	
Nishida T, 2000 [20]	0.47	0.33–0.63	
**Randomly pooled ES**	**0.60**	**0.34–0.85**	***p* < 0.001**
**180 to 210 days**			
Hilfrich J, 1975 [21]	0.79	0.57–0.91	94.86
Xiu Ying Yang, 2021 [13]	0.72	0.58–0.83	
Xiu Ying Yang, 2021 [13]	0.72	0.58–0.83	
Xiu Ying Yang, 2021 [13]	0.72	0.58–0.83	
Xiu Ying Yang, 2021 [13]	0.18	0.10–0.30	
Xiu Ying Yang, 2021 [13]	0.17	0.09–0.30	
Xiu Ying Yang, 2021 [13]	0.16	0.09–0.28	
Xiu Ying Yang, 2021 [13]	0.24	0.14–0.37	
Xiu Ying Yang, 2021 [13]	0.23	0.13–0.36	
Xiu Ying Yang, 2021 [13]	0.21	0.12–0.33	
**Randomly pooled ES**	**0.41**	**0.25–0.57**	***p* < 0.001**
**210 to 500 days**			
Marion SL, 2013 [14]	0.04	0.01–0.19	96.74
Marion SL, 2013 [14]	0.57	0.39–0.73	
Huang Y, 2012 [19]	0.75	0.67–0.82	
Huang Y, 2012 [19]	0.46	0.36–0.57	
Taguchi O, 1988 [23]	0.33	0.15–0.58	
Tunca JC, 1985 [11]	0.10	0.03–0.25	
Howell JS, 1954 [24]	0.60	0.48–0.71	
Howell JS, 1954 [24]	0.60	0.48–0.71	
Howell JS, 1954 [24]	0.60	0.48–0.71	
**Randomly pooled ES**	**0.45**	**0.25–0.65**	***p* < 0.001**
**Overall**			
**Randomly pooled ES**	**0.41**	**0.31–0.51**	***p* < 0.001**

**Table 6 biology-14-00073-t006:** Meta-analysis of the mortality rate of rats with DMBA-induced ovarian cancer grouped by treatment dose.

Study	ES	95% CI	Heterogeneity (%)
**0.5 mg–0.9 mg**			
Marion SL, 2013 [14]	0.06	0.03–0.12	-
Marion SL, 2013 [14]	0.06	0.03–0.12	
Xiu Ying Yang, 2021 [13]	0.18	0.1–0.3	
**Randomly pooled ES**	**0.08**	**0.03–0.13**	***p* < 0.001**
**1.0 mg–1.9 mg**			
Tunca JC, 1985 [11]	0.30	0.11–0.6	73.41
Xiu Ying Yang, 2021 [13]	0.46	0.33–0.6	
Xiu Ying Yang, 2021 [13]	0.15	0.08–0.28	
Xiu Ying Yang, 2021 [13]	0.20	0.12–0.32	
Xiu Ying Yang, 2021 [13]	0.14	0.07–0.26	
**Randomly pooled ES**	**0.21**	**0.11–0.31**	***p* < 0.001**
**2.0 mg–3.0 mg**			
Cai SQ, 2019 [8]	0.10	0.03–0.26	97.47
Cai SQ, 2019 [8]	0.20	0.10–0.37	
Cai SQ, 2019 [8]	0.07	0.02–0.21	
Cai SQ, 2019 [8]	0.03	0.01–0.17	
Cai SQ, 2019 [8]	0.03	0.01–0.17	
Ki Kwon Kim, 2003 [9]	0.33	0.18–0.53	
Taguchi O, 1988 [23]	0.46	0.3–0.64	
Hilfrich J, 1975 [21]	0.53	0.32–0.73	
Xiu Ying Yang, 2021 [13]	0.86	0.74–0.93	
Xiu Ying Yang, 2021 [13]	0.86	0.74–0.93	
Xiu Ying Yang, 2021 [13]	0.21	0.12–0.33	
Xiu Ying Yang, 2021 [13]	0.34	0.24–0.47	
**Randomly pooled ES**	**0.33**	**0.15–0.52**	***p* < 0.001**
**Overall**			
**Randomly pooled ES**	**0.27**	**0.16–0.37**	***p* < 0.001**

**Table 7 biology-14-00073-t007:** Meta-analysis of the mortality rate of rats with DMBA-induced ovarian cancer grouped by administration route.

Study	ES	95% CI	Heterogeneity (%)
**Soaked cotton**			
Cai SQ, 2019 [8]	0.10	0.03–0.26	17.68
Cai SQ, 2019 [8]	0.10	0.03–0.26	
Cai SQ, 2019 [8]	0.20	0.10–0.37	
Cai SQ, 2019 [8]	0.07	0.02–0.21	
Cai SQ, 2019 [8]	0.03	0.01–0.17	
Cai SQ, 2019 [8]	0.03	0.01–0.17	
**Randomly pooled ES**	**0.07**	**0.03–0.11**	***p* < 0.001**
**Peritoneal injection**			
Marion SL, 2013 [14]	0.06	0.03–0.12	79.00
Marion SL, 2013 [14]	0.06	0.03–0.12	
Tunca JC, 1985 [11]	0.30	0.11–0.60	
Hilfrich J, 1975 [21]	0.53	0.32–0.73	
Xiu Ying Yang, 2021 [13]	0.18	0.10–0.30	
Xiu Ying Yang, 2021 [13]	0.15	0.08–0.28	
Xiu Ying Yang, 2021 [13]	0.20	0.12–0.32	
**Randomly pooled ES**	**0.16**	**0.09–0.23**	***p* < 0.001**
**Saoked gauze**			
Ki Kwon Kim, 2003 [9]	0.33	0.18–0.53	96.98
Xiu Ying Yang, 2021 [13]	0.46	0.33–0.60	
Xiu Ying Yang, 2021 [13]	0.86	0.74–0.93	
Xiu Ying Yang, 2021 [13]	0.86	0.74–0.93	
Xiu Ying Yang, 2021 [13]	0.14	0.07–0.26	
Xiu Ying Yang, 2021 [13]	0.21	0.12–0.33	
Xiu Ying Yang, 2021 [13]	0.34	0.24–0.47	
**Randomly pooled ES**	**0.46**	**0.21–0.71**	***p* < 0.001**
**Gastric gavage**			
Taguchi O, 1988 [23]	0.46	0.30–0.64	-
**Randomly pooled ES**	**0.46**	**0.30–0.64**	***p* < 0.001**
**Overall**			
**Randomly pooled ES**	**0.27**	**0.16–0.37**	***p* < 0.001**

**Table 8 biology-14-00073-t008:** Meta-analysis of the mortality rate of rats with DMBA-induced ovarian cancer grouped by rat strains.

Study	ES	95% CI	Heterogeneity (%)
**SD**			
Cai SQ, 2019 [8]	0.10	0.03–0.26	96.43
Cai SQ, 2019 [8]	0.10	0.03–0.26	
Cai SQ, 2019 [8]	0.20	0.10–0.37	
Cai SQ, 2019 [8]	0.07	0.02–0.21	
Cai SQ, 2019 [8]	0.03	0.01–0.17	
Cai SQ, 2019 [8]	0.03	0.01–0.17	
Ki Kwon Kim, 2003 [9]	0.33	0.18–0.53	
Tunca JC, 1985 [11]	0.30	0.11–0.60	
Xiu Ying Yang, 2021 [13]	0.46	0.33–0.60	
Xiu Ying Yang, 2021 [13]	0.86	0.74–0.93	
Xiu Ying Yang, 2021 [13]	0.86	0.74–0.93	
Xiu Ying Yang, 2021 [13]	0.18	0.10–0.30	
Xiu Ying Yang, 2021 [13]	0.15	0.08–0.28	
Xiu Ying Yang, 2021 [13]	0.20	0.12–0.32	
Xiu Ying Yang, 2021 [13]	0.14	0.07–0.26	
Xiu Ying Yang, 2021 [13]	0.21	0.12–0.33	
Xiu Ying Yang, 2021 [13]	0.34	0.24–0.47	
**Randomly pooled ES**	**0.27**	**0.14–0.40**	***p* < 0.001**
**B6C3F**			
Marion SL, 2013 [14]	0.06	0.03–0.12	-
Marion SL, 2013 [14]	0.06	0.03–0.12	
**Randomly pooled ES**	**0.06**	**0.03–0.09**	***p* < 0.001**
**C3**			
Taguchi O, 1988 [23]	0.46	0.3–0.64	-
Hilfrich J, 1975 [21]	0.53	0.32–0.73	
**Randomly pooled ES**	**0.49**	**0.35–0.63**	***p* < 0.001**
**Overall**			
**Randomly pooled ES**	**0.27**	**0.16–0.37**	***p* < 0.001**

**Table 9 biology-14-00073-t009:** Meta-analysis of the mortality rate of rats with DMBA-induced ovarian cancer grouped by induction time.

Study	ES	95% CI	Heterogeneity (%)
**Up to 110 days**			
Cai SQ, 2019 [8]	0.10	0.03–0.26	27.27
Cai SQ, 2019 [8]	0.20	0.10–0.37	
Cai SQ, 2019 [8]	0.07	0.02–0.21	
Cai SQ, 2019 [8]	0.03	0.01–0.17	
Cai SQ, 2019 [8]	0.03	0.01–0.17	
**Randomly pooled ES**	**0.06**	**0.02–0.11**	***p* < 0.001**
**110 to 180 days**			
Cai SQ, 2019 [8]	0.10	0.03–0.26	-
Ki Kwon Kim, 2003 [9]	0.33	0.18–0.53	
**Randomly pooled ES**	**0.16**	**0.06–0.25**	***p* < 0.001**
**180 to 210 days**			
Hilfrich J, 1975 [21]	0.53	0.32–0.73	96.71
Xiu Ying Yang, 2021 [13]	0.46	0.33–0.60	
Xiu Ying Yang, 2021 [13]	0.86	0.74–0.93	
Xiu Ying Yang, 2021 [13]	0.86	0.74–0.93	
Xiu Ying Yang, 2021 [13]	0.18	0.10–0.30	
Xiu Ying Yang, 2021 [13]	0.15	0.08–0.28	
Xiu Ying Yang, 2021 [13]	0.20	0.12–0.32	
Xiu Ying Yang, 2021 [13]	0.14	0.07–0.26	
Xiu Ying Yang, 2021 [13]	0.21	0.12–0.33	
Xiu Ying Yang, 2021 [13]	0.34	0.24–0.47	
**Randomly pooled ES**	**0.39**	**0.20–0.58**	***p* < 0.001**
**210 to 500 days**			
Marion SL, 2013 [14]	0.06	0.03–0.12	85.25
Marion SL, 2013 [14]	0.06	0.03–0.12	
Taguchi O, 1988 [23]	0.46	0.3–0.64	
Tunca JC, 1985 [11]	0.30	0.11–0.6	
Howell JS, 1954 [24]	(Excluded)		
Howell JS, 1954 [24]	(Excluded)		
Howell JS, 1954 [24]	(Excluded)		
**Randomly pooled ES**	**0.16**	**0.05–0.26**	***p* < 0.001**
**Overall**			
**Randomly pooled ES**	**0.27**	**0.16–0.37**	***p* < 0.001**

## Data Availability

Data are contained within the article and Appendix A.

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
