# Peer review of "7,12-Dimethylbenz(a)anthracene as a Model for Ovarian Cancer Induction in Rats"

_biology, 2025, doi:10.3390/biology14010073_

Round 1
Reviewer 1 Report
Comments and Suggestions for Authors
The manuscript entitled, “DMBA as model for ovarian cancer induction in rat” is not worth acceptance for publication in biology,
Please describe the result section properly and avoid saying ‘See Figure A and Table B’.
It would be better to describe more in detail when there are several data from one manuscript. For instance, Cai SQ, 2019, Xiu Ying Yang, 2021 and any other paper were referred to in the same table with different ES or even the same ES, it cannot be understood without explanation.
Comments on the Quality of English Language
Please read through your manuscript because the template sentence remains. Also, check for proper English editing services as well.
Author Response
Dear Editor and Reviewers,
Thank you very much for your thoughtful comments and recommendations, which have significantly contributed to the enhancement of our work. We have revised the manuscript accordingly and believe that it has been substantially improved as a result. All additional text in response to your comments is highlighted in red in the revised manuscript. Additionally, we submitted the manuscript to MDPI’s English editing service. A detailed, point-by-point response to your comments is provided below.
Reviewer 1
Comment 1: Please describe the results section more clearly and avoid phrases like “See Figure A and Table B.”
Author’s Response 1: We have made the necessary revisions in the manuscript. Thank you for your suggestion.
Comment 2: It would be helpful to provide more detailed descriptions when multiple data points from a single study are referenced. For example, the studies by Cai SQ (2019), Xiu Ying Yang (2021), and others are referred to in the same table with different effect sizes (ES) or even the same ES. This needs further explanation.
Author’s Response 2: Thank you for your valuable comment. We have revised this section to clarify the references and ensure better understanding. All modifications are marked in yellow.
Comment 3: Please review the manuscript for template sentences that remain. Also, ensure proper English editing services are applied.
Author’s Response 3: Thank you again for your comment. We have submitted the manuscript to MDPI’s English editing service, and we hope that the revised version meets your expectations.
Reviewer 2 Report
Comments and Suggestions for Authors
In the manuscript “DMBA as model for ovarian cancer induction in rat” the Authors aimed to summarize the current knowledge of 7,12-Dimethylbenz(a)anthracene (DMBA) as inducer of ovarian cancer in rat. This systematic review clearly describes the flowchart of the search for and selection of articles conducted from the authors. Figures and tables are exhaustive. Since there is no systematic review in the literature that summarizes the standardized the chemical induction of ovarian carcinogenesis in rat model, the topic is very interesting.
However, I have some minor comments to share with the Authors:
1. Abstract writing should be improved in the part that concerns the description of the aims (lines 32-35).
2. I suggest checking the initial part of the Discussion section (lines 233-236); this part should probably be eliminated as it is part of the template.
3. In all figures, the Authors should clarify the choice of dotted lines of two different colors (black and red). In this way all figures should be more understandable.
Author Response
Dear Editor and Reviewers,
Thank you very much for your thoughtful comments and recommendations, which have significantly contributed to the enhancement of our work. We have revised the manuscript accordingly and believe that it has been substantially improved as a result. All additional text in response to your comments is highlighted in red in the revised manuscript. Additionally, we submitted the manuscript to MDPI’s English editing service. A detailed, point-by-point response to your comments is provided below.
Reviewer 2
Comment 1: The abstract needs improvement, particularly in the description of the aims (lines 32-35).
Author’s Response 1: Thank you for your comments and suggestions. We have revised this section to improve clarity and focus.
Comment 2: I suggest reviewing the initial part of the Discussion section (lines 233-236); this seems to be part of the template and should probably be eliminated.
Author’s Response 2: Thank you for your suggestion. We have removed this section as advised.
Comment 3: In all figures, the authors should clarify the choice of dotted lines of two different colors (black and red). This will make the figures easier to understand.
Author’s Response 3: Thank you. We would like to clarify that the color choices (black and red) were made by the software used for generating the figures. However, we have worked to improve the explanations in the figure legends for better clarity.
Reviewer 3 Report
Comments and Suggestions for Authors
This manuscript deals with a model for ovarian cancer induction in rats represented by the use of DMBA. This compound is known to induce carcinogenesis,
The work is well performed. Indeed, the authors analyzed in detail the different parameters that characterize this model. In particular, the amount of DMBA used, the way of induction of the tumor, the strains used, the time of appearance of the tumors, and the mortality rate.
The Authors concluded that DMBA can induce ovarian cancer. The doses Between 0.5 to 3 mg of DMBA can be applied depending on the route of treatment. The results can appear before 110 days, induction time should range between 110 and 500 days. More importantly, the number of animals used should be calculated for a mortality rate between 16% and 37%.
I think that this work merits to be published. However, I would add some discussion on the possibility of getting a standardization of the method as well as the tumors induced usually hit only epithelial ovary cells or other cell types present in the target organ,
This reviewer is not an expert on this specific way of inducing cancers, but I think it is important to discuss some points by looking at the data shown.
1- It appears that several parameters can be changed without giving so relevant differences. Is it possible to make a consensus to define the best experimental conditions?
2- How much this way to induce cancer is similar to what can happen in humans?
3- Are the molecular alterations found in the plethora of experiments performed and reported similar or not?
4- How many of these molecular alterations are indeed present in human tumors?
5- How much (if any) does this model resemble the events usually present in ovarian cancer?
6- Is it possible to compare this way of generating ovarian cancer and others published in the literature?
7- Is this model reasonable to generate ovarian cancer, considering the potential heterogeneity of mutations in human patients?
8- Does this model take into account the immune response of the host?
This point is extremely relevant in the era of immune check point blockers.
Author Response
Dear Editor and Reviewers,
Thank you very much for your thoughtful comments and recommendations, which have significantly contributed to the enhancement of our work. We have revised the manuscript accordingly and believe that it has been substantially improved as a result. All additional text in response to your comments is highlighted in red in the revised manuscript. Additionally, we submitted the manuscript to MDPI’s English editing service. A detailed, point-by-point response to your comments is provided below.
Reviewer 3
Comment 1: It appears that several parameters can be altered without resulting in significant differences. Is it possible to reach a consensus on the best experimental conditions?
Author’s Response 1: Thank you for your comments. As discussed in our manuscript, several factors can influence the experimental results. However, based on the success of tumor induction, the best experimental conditions for studying ovarian cancer in rats were established by Kim KK and colleagues in 2003.
Comment 2: How similar is this method of cancer induction to what occurs in humans?
Author’s Response 2: Thank you. Unfortunately, we cannot yet compare these two processes directly, as the mechanisms of ovarian carcinogenesis in humans are still not fully understood. This was not the focus of our manuscript. We aimed to summarize existing literature to assist researchers in selecting an appropriate model for ovarian cancer studies. We hope our work contributes to this area.
Comment 3: Are the molecular alterations found in the experiments described similar or not?
Author’s Response 3: Thank you. Unfortunately, our manuscript does not focus on molecular alterations, as we were primarily concerned with identifying the optimal conditions (timing, route, and doses of DMBA) for ovarian cancer induction in rats. We apologize for not addressing this in more detail.
Comment 4: How many of these molecular alterations are present in human tumors?
Author’s Response 4: Thank you. While we can hypothesize that certain molecular pathways involved in ovarian carcinogenesis may be similar in rats and humans, our manuscript does not delve into this comparison. We are currently preparing another study that will address this aspect in more detail.
Comment 5: How similar is this model to the events typically observed in ovarian cancer?
Author’s Response 5: Thank you. In this manuscript, our focus was on the availability and applicability of the model. We are currently working on a follow-up study that will focus on the molecular aspects of the DMBA ovarian cancer model and compare them with human ovarian cancer.
Comment 6: Can this method of generating ovarian cancer be compared with other published methods?
Author’s Response 6: Thank you. Yes, based on the histological analysis of tumors, we can confirm that this model induces tumors similar to those seen in other models. However, researchers must evaluate each model based on their specific research objectives.
Comment 7: Is this model suitable for generating ovarian cancer, considering the potential heterogeneity of mutations in human patients?
Author’s Response 7: Thank you. This model is useful for various pharmacological and functional studies. However, as with all tumor models, human cancer is heterogeneous, and while similar pathways may be involved, this model is not an exact biomimetic system.
Comment 8: Does this model take into account the immune response of the host? This is particularly relevant in the era of immune checkpoint blockers.
Author’s Response 8: Thank you. Although we did not specifically address the immune response in this study, we acknowledge that it plays a critical role in tumor development. Other studies have explored this aspect in depth, and we hope to consider this in future research.